# Diversity of Potential Resistance Mechanisms in Honey Bees (*Apis mellifera*) Selected for Low Population Growth of the Parasitic Mite, *Varroa destructor*

**DOI:** 10.3390/insects16040385

**Published:** 2025-04-04

**Authors:** Alvaro De la Mora, Paul H. Goodwin, Nuria Morfin, Tatiana Petukhova, Ernesto Guzman-Novoa

**Affiliations:** 1Department of Veterinary Pathology, Western College of Veterinary Medicine, University of Saskatchewan, 52 Campus Drive, Saskatoon, SK S7N 5B4, Canada; 2School of Environmental Sciences, University of Guelph, 50 Stone Road East, Guelph, ON N1G 2W1, Canada; pgoodwin@uoguelph.ca (P.H.G.); eguzman@uoguelph.ca (E.G.-N.); 3Department of Entomology, Faculty of Agricultural and Food Sciences, University of Manitoba, 12 Dafoe Road, Winnipeg, MB R3T 2N2, Canada; nuria.morfin@umanitoba.ca; 4Department of Population Medicine, University of Guelph, 50 Stone Road East, Guelph, ON N1G 2W1, Canada; tpetukho@uoguelph.ca

**Keywords:** *Apis mellifera*, *Varroa destructor*, selective breeding, resistance mechanisms, colony collapse disorder, immunity, deformed wing virus

## Abstract

The parasitic mite *Varroa destructor* is the most damaging biotic stressor of honey bees (*Apis mellifera*) worldwide. Breeding bees for resistance against *V. destructor* is a sustainable long-term approach to managing the mite. This study analyzed several mechanisms of resistance of honey bees that were bred for low (resistant) and high (susceptible) *Varroa* population growth (LVG and HVG, respectively). After three generations of selection based on mite fall, LVG bees had significantly higher immunity detected as stronger behavioral (hygienic and grooming behaviors), cellular (haemocyte concentration), and humoral (hymenoptaecin 2 and defensin 2 antimicrobial peptide gene expression and lower DWV levels) immunity compared to HVG bees. These results indicate that selecting bees for LVG indirectly selects for bees with multiple resistance mechanisms against *V. destructor*. Thus, it appears that the LVG trait is associated with multiple genes.

## 1. Introduction

*Varroa destructor* is a parasitic mite of the Western honey bee (*Apis mellifera* L.) that has been linked to colony losses worldwide, mainly in countries in the Northern hemisphere [1]. *Varroa* feeds upon the fat body tissue and haemolymph of honey bees [2,3], suppressing their immune system and shortening their lifespan [4,5,6]. A major factor in the detrimental effects of *Varroa* is its role as a vector of bee viruses, including deformed wing virus (DWV) [7], which reduces the longevity of adult bees and causes a deformed body and wings in developing bees [8,9]. Hence, both *V. destructor* and DWV are linked to honey bee colony losses.

One control strategy is breeding honey bees for *Varroa* resistance [10]. For example, De la Mora et al. [11,12] carried out three generations of bidirectional selection for low (resistant) and high (susceptible) *Varroa* population growth (LVG and HVG, respectively) based on mite fall in colonies at two different time points. The LVG genotype demonstrated lower *Varroa* population growth and levels, as well as lower colony DWV levels and winter colony mortality than the HVG genotype. However, the mechanisms involved in low *Varroa* population growth were not reported. One possible mechanism is grooming behavior, in which bees bite and dislodge mites from their bodies [13]. Variation between bee genotypes has been demonstrated for the number of visibly damaged *Varroa* among the fallen mites, presumably due to grooming behavior [14,15,16]. This trait has been used for selecting honey bee stocks, such as the ‘mite-biter’ [17]. Another possible resistance mechanism is hygienic behavior, in which bees identify and remove not only diseased or dead broods from their cells, but also *Varroa*-infested broods [10,18,19]. Honey bee stocks have been selected for this trait, such as the *Varroa*-sensitive hygiene stock that was selected for high detection and removal of mite-infested broods [20,21].

There are other mechanisms possible for *Varroa* resistance in addition to behavioral responses. One is an enhanced cellular immunity, such as increased haemocytes, and another is higher humoral immunity, such as increased defense compounds [22]. Haemocytes are immune cells that engulf pathogens present in the bees’ haemolymph [23]. They also promote wound healing and haemolymph clotting after the *V. destructor* attack [24] and produce anti-viral compounds [25,26]. Humoral defense compounds include antimicrobial peptides (AMPs) that neutralize pathogenic microorganisms, including viruses [1,27]. An example of antiviral immunity is the trait called ‘Suppressed in ovo virus infection (SOV)’ that is found in resilient colonies to DWV infections [28]. The primary mechanism of insect antiviral defense is RNA interference (RNAi), and high virus levels in honey bees can suppress key RNAi components [29].

While the LVG genotype clearly showed resistance to *Varroa* compared to the HVG genotype with three rounds of selection for low rates of *Varroa* population growth [12], the mechanisms were not examined. The current study examined LVG and HVG bees for several potential mechanisms of *Varroa* resistance, including behavioral resistance (grooming and hygienic behavior), cellular immunity (haemocyte concentrations), humoral immunity (expression of the AMP genes, defensin 2 (*AmDef-2*), and hymenoptaecin 2 (*AmHym-2*)). Additionally, *Varroa* parasitized bees of the two genotypes were assessed for DWV levels to determine if antiviral resistance was involved for bees individually parasitized by the mite. Therefore, this study provides a better understanding of whether selection for LVG involves one or more mite-resistance mechanisms.

## 2. Materials and Methods

### 2.1. Location and Genotype Selection

Experiments were conducted at the Honey Bee Research Center (HBRC), University of Guelph, Guelph, ON, Canada (43.5448° N, 80.2482° W). Honey bees underwent three generations of selection for either low (resistant) or high (susceptible) *Varroa* population growth based on mite fall in colonies at two different time points to obtain the LVG and HVG genotypes, respectively [11,12]. Each generation was one year and *Varroa* treatments were applied in colonies in the Fall of each year using amitraz (Apivar, Veto-Pharma, Saint-Benoit-du-Sault, 36310 Chaillac, France) following the manufacturer’s instructions.

### 2.2. Hygienic and Grooming Behaviors at the Colony Level

Seventeen randomly selected colonies per genotype were tested on each generation in late May to assess hygienic behavior by freeze-sacrificing capped worker brood with liquid nitrogen as per Spivak and Reuter [30]. Briefly, a metal cylinder (4.10 cm internal diameter) was placed on four sections of comb containing ~140 capped cells (~35 capped brood cells/section) from each of the tested colonies. Then, 250 mL of liquid nitrogen was poured into its interior to kill the brood. The number of cells uncapped and cleaned by the bees was counted 24 h after freezing the brood to calculate the percentages of uncapped and cleaned cells.

Twenty randomly selected colonies per genotype were assessed for each generation in late August for grooming behavior by calculating the percentage of damaged mites with body dents and/or mutilated legs out of the total number of mites collected from sticky boards [17,31]. The mites were transferred with a fine paint brush from a colony sticky board into a Petri dish. To observe damage to mites, they were positioned with their ventral side facing up under a stereoscopic microscope (Olympus SD-ILK, Optical Co., Tokyo, Japan). The number of mites with mutilated legs and/or dents on the idiosoma were counted. The rate of mutilated mites was determined by dividing the number of damaged mites over the total number of mites analyzed.

### 2.3. Grooming Behavior at the Individual Level

Three randomly selected colonies per genotype of the third generation were sampled for adult bees in late August to assess self-grooming behavior in the laboratory [32]. A total of 1534 bees were assessed. Briefly, bees from brood-chamber frames were shaken into a 5 L container, then 500 mL of them was scooped into an open-mouth 1 L container with a 100 micron honey filter cloth (Dancing Bee, Port Hope, ON, Canada) as a lid. Each bee taken from the container was individually placed in a Petri dish (100 mm × 15 mm; Fisher Scientific, Mississauga, ON, Canada) that had a perforated plastic lid, and was allowed to acclimate for 2 min. Then, approx. 20 mg of wheat flour (Robin Hood, Markham, ON, Canada) was applied onto the thorax of the bee using a fine paint brush (6 mm × 11 mm; DeSerres, Oakville, ON, Canada) as an irritant to stimulate grooming instances. Wheat flour is a proxy of *V. destructor* for grooming behavior assays [32]. The time (s) taken by each bee to start performing the first grooming instance was measured for up to 3 min using a stopwatch with a resolution of 1/100 s and an accuracy of 0.001% (Catalog number 06-662-56. Fisherbrand, Mississauga, ON, Canada). This was performed as a blind test to the observer. The intensity of grooming (light or intense) was also recorded as per Guzman-Novoa et al. [15]. Bees were classified as ‘light’ groomers if they used no more than two legs and if slow motions to remove the irritant were observed, or as ‘intense’ groomers if they used three or more legs and if vigorous shaking and wiping motions to remove the irritant were observed. Bees showing intermediate expressions of grooming instances were discarded from the analysis.

### 2.4. Source of V. destructor

To obtain female *Varroa*, frames covered with adult bees from highly infested colonies that were unrelated to the experimental bees were shaken inside a plastic bag, and then the bees were anesthetized with CO_2_. The bees were transferred to a plastic chamber (30 × 18 × 4 cm) that was divided into upper and lower areas by a 3 mm metal mesh that served to support the anesthetized bees. The chamber was placed on an orbital shaker (Eberbach, Van Buren, MI, USA) set at 400 rpm for 10 min to dislodge mites attached to the bees [33]. Fallen mites were recovered from the lower area of the chamber and placed in a plastic Petri dish lined with a moist paper towel using a fine paint brush. The bees used for this procedure were returned to their original hive after recovering from anesthesia.

### 2.5. Cellular Immunity

Brood nest frames of three colonies of each of the LVG and HVG genotypes of the third generation of selection were shaken into a 5 L container, and 50 bees were individually transferred to five Benton wooden cages (75 × 25 × 16 mm) with 10 bees per cage. The bees in the cages were treated as follows: (1) HVG bees without *Varroa*; (2) HVG bees with *Varroa*, (3) LVG bees without *Varroa*; (4) LVG bees with *Varroa*. The experiment was replicated three times using a different source colony per replicate. For *Varroa* exposure, bees were artificially parasitized with one mite per individual by placing a mite with a fine paintbrush onto each bee through the cage screen, and the cages were maintained at 32–35 °C and 60% RH for eight days [4]. The bees were fed queen candy (65% sucrose syrup mixed with powdered sucrose as needed to obtain a soft mass) and watered twice a day by placing a clean 2 × 2 cm sponge piece (Scotch Brite, Two Harbors, MN, USA) with dsH_2_O on the cage screen. Samples of 10 bees of each genotype that had or had not been exposed to *V. destructor* were collected at eight days post-exposure to the mite. Each bee was collected directly from the cage and held manually with the index and thumb fingers to expose the dorsal part of the insect’s abdomen. Then, each bee was pierced between the third and fourth tergite with a #7 entomological pin (Bioquip, Rancho Dominguez, CA, USA), and a sample of 4 µL of haemolymph was collected with a 10 µL micropipette (Labnet International, Edison, NJ, USA).

The concentration of haemocytes/µL of haemolymph was measured as per Koleoglu et al. [5]. Briefly, the haemolymph sample was evenly spread over a microscope slide previously marked with two 1 × 1 cm squares, each divided into four 0.5 × 0.5 cm squares using a fine point permanent marker (Sharpie, Atlanta, GA, USA). The smears were allowed to airdry for 15 min and then fixed with 10 µL of 95% methanol. The smears were dyed with the Hema 3 Stat Pack kit (Fisher Scientific, Fair Lawn, NJ, USA) following the manufacturer’s instructions. After staining the smears, haemocytes were counted at 400× magnification under an optic microscope (Leica Microsystems, Wetzlar, Germany) equipped with a 10 × 10 mm ocular reticule having a 100 cells grid (2.5 μm^2^ each). Eight counts were performed with two areas selected in each 0.5 × 0.5 cm square, one near the center and the other near the corner of the 1 × 1 cm square. Haemocytes were counted in a zig-zag pattern starting at the top left-hand corner of each square. To calculate the number of haemocytes per μL of haemolymph in each sample, the following equation was used:No. haemocytes/µL = ((Ʃ haemocytes/8) × 1322)/4

The adjustment factor of 1322 was calculated from Murphy and Davidson [34], which corresponds to the number of microscope reticles. This procedure was repeated three times.

### 2.6. Humoral Immunity

Samples of 10 bees harvested from broodnest frames that were treated in cages as per the cellular immunity measurements were collected eight days after being infested and stored at −80 °C. Detection and quantification of gene expression was performed as per Morfin et al. [35], with modifications for RNA extraction. Briefly, bees were macerated with a pestle and mortar in 5000 µL of One Step RNA Reagent (BioBasic, Markham, ON, Canada), following instructions from the manufacturer. The macerate was transferred to a new 1.5 mL centrifuge tube and incubated for 5 min at 20–22 °C. Then, 300 µL of chloroform was added, and the tube was vortexed (Thermo Fisher, Waltham, MA, USA) at 7000 rpm for 15 s. After incubation at 20–22° for 2–3 min, the tube was centrifuged (VWR, Mississauga, ON, Canada) at 12,000× *g* for 15 min at 4 °C. The aqueous phase was transferred to a new 1.5 mL microcentrifuge tube, and 500 µL of 99% isopropanol was added. The tube was incubated at 20–22 °C for 10 min and centrifuged at 12,000× *g* for 10 min at 4 °C. The supernatant was discarded, and the pellet was washed by adding 1000 µL of 70% ethanol, mixing by inversion and followed by centrifugation at 7500× *g* for 5 min at 4 °C. This was repeated two additional times. The pellet was airdried at room temperature. Finally, the RNA pellet was dissolved in 30 µL of Invitrogen UltraPure H_2_O (Fisher Scientific, Waltham, MA, USA), and RNA quality was assessed using a NanoDrop Lite (Thermo Fisher, Waltham, MA, USA). The RNA samples were stored at −80 °C.

cDNA was prepared by using 2000 ng of RNA and the Fermentas Revert Aid H Minus First Strand cDNA Synthesis Kit (Fisher Scientific, Mississauga, ON, Canada), following the manufacturer’s instructions. Quantitative real-time PCR (RT-qPCR) was used to analyze the expression of the immune-related genes, defensin 2 (*AmDef-2*) and hymenoptaecin 2 (*AmHym-2*) [36]. For *AmDef-2*, a 20 µL reaction contained 2 µL cDNA, 10 µL PowerUp SYBRgreen (Supermix 2×) (Applied Biosystems, Foster City, CA, USA), 0.8 µL each of the 400 nM forward and reverse primers (Appendix A) [37], and 6.4 µL of nuclease free H_2_O. For *AmHym-2*, a 20 µL reaction contained the same reagents and volumes, except for 0.6 µL each of the 400 nM forward and reverse primers (Appendix A) [37] and 6.8 µL of nuclease-free H_2_O. The expression of those genes was measured relative to the expression of the constitutive gene 40S ribosomal protein S5 (*AmRPS5*) [36] with each reaction containing 2 µL of cDNA, 10 µL of Supermix 2×, 1.4 µL each of the 700 nM forward and reverse primers (Appendix A) [37], and 5.2 µL of nuclease-free H_2_O. The negative control included 2 µL of nuclease free H_2_O, and the positive control included 2 µL of the diluted synthetic 300 bp gBlock of the target gene (Integrated DNA Technologies, Coralville, IA, USA) (Appendix A) [35]. RT-qPCR was performed with a QuantStudio3 thermocycler (Real-Time PCR Systems, Fisher Scientific, Pittsburgh, PA, USA) with one cycle at 50 °C for 2 min, one cycle at 95 °C for 10 min, and 40 cycles at 95 °C for 10 s and 60 °C for 60 s. The expression level of the *AmRPS5* reference gene in each sample was used to normalize the expression level of the target gene using the 2^−ΔΔ^ (Livak) method [38] with the HVG genotype as calibrator. The QuantStudio3 real-time PCR detection system was used to calculate the normalized expression level.

### 2.7. DWV Infection Levels

Samples of 18–22 bees treated the same as per the cellular immunity measurements were collected at eight days post-treatment and stored at −80 °C. Detection and quantification of DWV was performed as per Morfin et al. [39], with modifications for RNA extraction as described above. cDNA was prepared with 2000 ng of RNA for each sample using the Fermentas Revert Aid H Minus First Strand cDNA Synthesis Kit (Fisher Scientific, Burlington, ON, Canada), following the manufacturer’s instructions. cDNA was amplified by RT-qPCR with a QuantStudio3 thermocycler (Real-Time PCR Systems, Fisher, Pittsburgh, PA, USA). Each qPCR reaction of the helicase gene of DWV type A consisted of 20 µL containing 2 µL of cDNA, 0.4 µL of both 200 nM forward and reverse primers (Appendix A) [40], 10 µL PowerUp^TM^ Sybergreen (2×) (Applied Biosystems, Foster City, CA, USA), and 7.2 µL nuclease-free H_2_O. The negative control was 2 µL of nuclease-free H_2_O instead of cDNA. The positive control included 2 µL of a 300 bp gBlock (Integrated DNA Technologies, Coralville, IA, USA) that included the sequences of the forward primer, amplicon, and reverse primer (Appendix A) [39]. PCR conditions consisted of one cycle at 48 °C for 15 min, one cycle at 95 °C for 10 min, 40 cycles at 95 °C for 15 s and 60 °C for 60 s, followed by one cycle at 68 °C for 7 min. Calibration curves to convert Ct values to DWV genome copies (gc) were carried out using 300 bp gBlocks (Integrated DNA Technologies, Coralville, IA, USA) (Appendix A) diluted in dsH2O to 10 ng/μL that was then used to make 10-fold serial dilutions from 10^9^ to 10^2^. Using a plot of Ct values versus DWV copy number (log10), a linear equation was used to calculate the DWV gc [41,42]. Three technical repetitions were performed for each qRT-PCR. Randomly selected amplicons of DWV PCR product were sequenced at the University of Guelph Laboratory Services to confirm identity.

### 2.8. Statistical Analyses

Data were analyzed for normality with the Shapiro–Wilk test. When not normally distributed, data were transformed. Data on DWV copies, haemocyte counts, gene expression, and time to start grooming were log-transformed. Data on percentage hygienic behavior were arcsine-square root transformed. The transformed data were subjected to analyses of variance and Fisher-protected LSD tests to compare means when significant differences were found. Data on percentages or proportions, including damaged mites, and grooming intensity between genotypes, were analyzed with contingency tables and corrected χ^2^ tests. Statistical analyses were performed with the R 4.1.1. software [43].

## 3. Results

### 3.1. Hygienic and Grooming Behaviors at the Colony Level

Hygienic behavior estimated from the percentage of frozen brood removed by bees of the LVG colonies was significantly higher than that of the HVG colonies in the first and second generations, but not in the third generation (F_1,97_ = 15.25, *p* < 0.001) (Figure 1a). There were no significant effects of genotype x generation interaction (*p* > 0.05). The greatest difference in frozen brood removal was in the first generation of LVG colonies, when it was 35% higher than that of the HVG colonies.

Grooming behavior measured by the mean percent dented and/or mutilated mites from LVG colonies was significantly higher than that from HVG colonies in the first and third generations (χ^2^ = 47.1, n = 1323, *p* < 0.0001; χ^2^ = 5.7, n = 230, *p* < 0.05, respectively), but not in the second generation (χ^2^ = 3.5, n = 965, *p* > 0.05) (Figure 1b). Mite damage rates from LVG colonies in the first and third generations were 61% and 30% greater than those of HVG colonies (Figure 1b), respectively, which was associated with 47% and 66% lower *Varroa* infestation levels in adult bees and brood, respectively, as reported previously [12]. The greatest difference was in the first generation, where mite damage was 42% in LVG compared to 16% in HVG colonies. There were no significant effects of genotype × generation interaction (*p* > 0.05). For both hygienic and grooming behaviors, there was no evidence that each round of selection progressively increased the traits, which appeared to differ by generation, possibly by random variation or environmental factors each year. Therefore, all subsequent examinations of traits of individual LVG and HVG bees were only performed on the third generation of selection.

### 3.2. Grooming Behavior at the Individual Level

The proportion of individuals performing intense grooming was significantly higher for bees from LVG colonies (0.42) than for bees from HVG colonies (0.36) (χ^2^ = 6.26, *p* < 0.05) (Figure 2a). However, there was a significantly shorter grooming starting time for intense but not light groomers with HVG-intense groomers being significantly slower than LVG-intense groomers, but not for light groomers (F_1,1532_ = 1.55, *p* < 0.0001) (Figure 2b). The fastest response was with LVG-intense groomers (4.32 s), followed by HVG-intense groomers (4.66 s), then LVG-light groomers (12.58 s), and the slowest was with HVG-light groomers (13.7 s). There were significant effects for genotype x intensity interaction (*p* < 0.05) with LVG bees that performed intense grooming starting to groom faster than HVG bees.

### 3.3. Cellular Immunity

Haemocyte concentration was significantly higher in the haemolymph of non-parasitized LVG bees (25.1 × 10^3^ haemocytes per µL) compared to non-parasitized HVG bees (11.1 × 10^3^ haemocytes per µL), as well as in the haemolymph of parasitized LVG bees (18.0 × 10^3^ haemocytes per µL) compared to parasitized HVG (3.5 × 10^3^ haemocytes per µL) bees (F_1,38_ = 5.0, *p* < 0.05) (Figure 3). Parasitism significantly lowered haemocyte concentration in HVG bees, but not in LVG bees. There were no significant interaction effects of genotype x treatment (*p* > 0.05).

### 3.4. Humoral Immunity

The relative expression (RE) of defensin 2 (*AmDef-2*) in LVG and HVG bees was significantly higher in non-parasitized LVG (4.33 RE) compared to non-parasitized HVG (0.77 RE) bees, as well as between parasitized LVG (11.02 RE) compared to parasitized HVG (4.08 RE) bees (F_1,20_ = 4.2672, *p* < 0.05; Figure 4a). However, there were no significant differences when comparing non-parasitized to parasitized bees within either the LVG or HVG genotypes (F_1,20_ = 0.7343, *p* > 0.05). There were significant interaction effects of genotype x treatment (*p* < 0.05).

Hymenoptaecin 2 (*AmHym-2*) relative expression in LVG and HVG bees was not significantly different between non-parasitized LVG (0.17 RE) and HVG (0.25) bees (F_1,22_ = 0.74, *p* > 0.05), but was significantly higher in parasitized LVG (4.84 RE) than HVG (0.46 RE) bees (F_1,22_ = 6.40, *p* < 0.05; Figure 4b). Comparing non-parasitized to parasitized bees within a genotype, there was significantly higher expression for parasitized LVG bees with a 4.67 RE increase, but not for parasitized HVG bees with only a 0.21 RE increase. There were significant interaction effects of genotype × treatment (*p* < 0.05).

### 3.5. DWV Infection Levels

DWV infection levels in LVG and HVG bees were significantly lower in non-parasitized LVG (4.63 copies × 10^6^ per µg RNA) than non-parasitized HVG (5.77 copies × 10^6^ per µg RNA) bees, as well as significantly lower in parasitized LVG (6.19 copies × 10^6^ per µg RNA) than parasitized HVG (8.31 copies × 10^6^ per µg RNA) bees (F_3,92_ = 5.5288, *p* < 0.05; Figure 5). Parasitism resulted in higher DWV levels for both LVG and HVG bees, but the increase was only 1.56 versus 2.54 copies × 10^6^ per µg RNA in LVG and HVG bees, respectively. There were no significant interaction effects of genotype x treatment (*p* > 0.05).

## 4. Discussion

Although it was reported that selection of colonies for high or low *Varroa* population growth resulted in LVG colonies with almost 90% less *Varroa* growth than HVG colonies and significantly lower *Varroa* infestation levels by the third generation, the mechanisms of resistance were not examined [12]. One mechanism of resistance to *Varroa* is hygienic behavior. Increased resistance to *Varroa* has been shown to result from increased hygienic behavior where bees detect *Varroa* infested broods in cells and then remove them [10,30,44,45]. Low *Varroa* infestation levels in colonies have also been linked to a specialized type of hygienic behavior, *Varroa* sensitive hygiene (VSH), in which bees are also able to identify the cells where the mite is reproducing, restraining *Varroa* reproduction and population growth [18,20,46]. This behavior has been observed at high frequencies in honey bee colonies selected for VSH [20].

The rate of hygienic behavior of LVG colonies was significantly higher than that of HVG colonies in the first two generations but not in the third generation with the difference between LVG and HVG colonies becoming progressively less with each generation. It appears that this trait may have been indirectly selected initially acting as one contributing factor to resistance but possibly becoming less of a contributor with each generation. In a previous study selecting for low *Varroa* growth, greater hygienic behavior was not found to be related to *Varroa* resistance [47]. However, directly selecting for increased hygienic behavior reduces *Varroa* populations in honey bee colonies, with bees removing 72% more killed brood and resulting in 60% lower mite infestation [48]. Other examples selecting for hygienic behavior resulted in 78% [21] and 52% lower mite infestation levels [46]. The inconsistent relationship between an increase in hygienic behavior with low *Varroa* population growth in this study indicates that it is only one contributor to resistance and other mechanisms may be involved.

*Varroa* resistance in honey bees has also been reported to result from increased grooming behavior where bees remove *Varroa* from their own bodies or that of other bees [1]. Grooming behavior at the colony level was indirectly assessed by analyzing the percentage of damaged (dented and/or mutilated) mites recovered from colonies [17]. Mite damage rate in LVG colonies was significantly higher than that in HVG colonies in the first and third generations, but not in the second generation, and thus, it was not consistently associated with low *Varroa* growth. For previous studies selecting for LVG that examined grooming behavior, Rinderer et al. [49] found 33% more damaged mites associated with 73% lower *Varroa* infestation levels, whereas Lodesani et al. [50] found that mite damage varied by generation with no correlation between damaged mite numbers and lower *Varroa* infestation levels. However, direct selection for increased grooming behavior can provide effective *Varroa* resistance, such as 70% more damaged mites with 34% lower *Varroa* infestation levels [16], 64% more damaged mites and 55% lower *Varroa* infestation levels [51], and 43% more damaged mites and 64% lower *Varroa* infestation levels [52]. Like hygienic behavior, grooming behavior appears to contribute to resistance as it was associated with the LVG genotype in two of the three generations of selection but not in the second generation. Therefore, additional mechanisms must be involved in the resistance to *Varroa* of the LVG genotype.

One reason that grooming behavior was not consistently linked with low *Varroa* population growth in all three generations of this study could be that grooming behavior is a polygenic trait, as indicated by nine protein markers being associated with it [53]. Thus, it is possible that different sets of grooming behavior genes may have been more strongly selected in certain generations. This may also explain why there is a considerable range in the relationship between the level of damaged mites and infestation levels reported between different studies. It is also possible that a proportion of the mutilated and/or dented mites collected from the colonies were also a result of VSH or uncapping/recapping behavior as it has been demonstrated that bees injure mites when they remove them from cells [54]. This study did not distinguish between the ways that *Varroa* could have been injured and so the numbers could reflect both VSH and grooming behavior. Future studies on LVG and HVG genotypes are warranted to investigate the relative contribution of each of these behaviors to *Varroa* damage.

Grooming behavior was also assessed at the individual level by measuring grooming intensity and grooming starting time in the third generation. LVG bees had a significantly greater proportion of intense groomers relative to light groomers, and intense-grooming LVG bees started grooming sooner than intense- and light-grooming HVG bees. A higher proportion of intense groomers can contribute to *Varroa* resistance as intense groomers can dislodge over 80% of mites from their bodies compared to light groomers [15,55]. These results agree with the significantly greater number of damaged mites collected from LVG colonies in comparison with HVG colonies. None of the previous studies selecting for LVG examined grooming starting time or grooming intensity [11,12,14,47,49,50,56]. In this study, 14% more LVG bees performed grooming events within 3 min of a stimulus than HVG bees. By comparison, studies directly selecting genotypes for increased grooming behavior resulted in 42% [47] and 32% [16] more bees performing grooming events within 3 min of a stimulus compared to non-selected genotypes. The higher grooming intensity and faster grooming starting time in the third generation of LVG bees supports grooming behavior as one of the multiple mechanisms contributing to the LVG phenotype. Studies conducted in Europe did not find a correlation between mite fall, damaged mites, or hygienic behavior, and the survival of *Varroa*-infested colonies [14]. However, other studies conducted in the Americas and the Middle East have shown a relationship between these traits and colony health and survivorship [16,17,31,45]. Perhaps the relative contribution of different traits to the resistance of bees to *Varroa* and their effect on colony survivorship varies depending on the environment or some other factor. Therefore, more studies conducted in different regions of the world are needed to determine whether these traits are useful to increase colony survivorship in different environments.

One potential behavioral mechanism of resistance that was not assessed in this study was recapping, which involves young bees detecting infested brood, opening the cells, occasionally removing the mites, and re-sealing the cells [57]. This behavior may reduce mite fertility and fecundity because mite reproduction and mortality of mite offspring is higher in recapped cells than in mite-infested cells that are not opened [58]. Recapping could be an important mechanism of resistance against *V. destructor* in some honey bee populations. For example, a population of honey bees that has survived *Varroa* infestations without acaricide treatments for more than 20 years in Cuba had high rates of recapping behavior of mite-infested cells, with the removal of over 80% of mites from parasitized cells [59]. Thus, artificially parasitizing capped broods of HVG and LVG colonies to assess recapping would be useful to determine if recapping is increased in LVG colonies. The contribution of recapping to *Varroa* resistance in honey bee colonies warrants further investigation before incorporating this trait into selective breeding programs.

Another potential behavioral mechanism of resistance that was not assessed in this study was mite non-reproduction (MNR) or suppressed mite reproduction (SMR), in which a percentage of *Varroa* foundress females do not reproduce or have low fecundity in parasitized brood [60,61]. MNR and SMR have been shown in honey bee populations demonstrating resistance to *Varroa* [62,63,64]. Additionally, low mite fecundity is consistently found in presumed *Varroa*-resistant populations of honey bees [64,65,66]. The mechanisms that cause MNR are not clear, but they may be behavioral, as VSH is responsible for a significant proportion of low mite fertility, as hygienic bees remove reproducing mites from capped cells [18,46]. The value of this trait was evidenced during an eight-year breeding program that selected for a bee genotype with lower mite reproduction, and that resulted in a 60% reduction in *Varroa* populations [60]. It is possible that the MNR and recapping traits had been indirectly selected by the overall trait of LVG, which remains to be investigated.

Cellular immunity, based on the concentration of haemocytes in the haemolymph, was significantly higher in LVG than in HVG bees both with and without *Varroa* parasitism. As increased cellular immunity has never been related to *Varroa* resistance; this was unexpected. Because newly emerged bees were determined to not have *Varroa* parasitism prior to being placed in cages in this study, it could be proposed that increased haemocyte concentrations in bees without *Varroa* shows that a basal function has been indirectly selected during the development of LVG colonies. Having higher concentrations of haemocytes in LVG bees free of pathogens and parasites could benefit bees because haemocytes in insects are involved in many processes, including molting and development, hypoxia survival, vitellogenin production, iron transport, lipoprotein synthesis, apoptotic cell clearing, and other non-immune activities [67]. Thus, many beneficial traits may have been increased in LVG bees due to higher haemocyte concentrations. In addition, only the decline in haemocyte concentration with *Varroa* parasitism was significant in HVG bees. Like non-parasitized bees, parasitized LVG bees had more haemocytes than parasitized HVG bees, which could contribute to resistance by allowing them to reduce *Varroa* feeding as haemocytes are involved in the production of reactive oxygen and nitrogen species, which can be toxic [25]. In addition, more haemocytes in LVG bees could increase clotting and wound healing, including wounds created by *Varroa* when piercing the bee exoskeleton [24], and more haemocytes could participate in humoral immune pathways for AMP production [68]. Thus, increased haemocytes could be contributing to resistance in multiple ways. It is unknown if the higher haemocyte concentrations in healthy LVG bees were due to greater production of haemocytes during development, greater storage of haemocytes, or a longer haemocyte lifespan, but higher concentrations could allow LVG bees to be better able to compensate for the loss of haemocytes during *Varroa* parasitism. Fruit fly resistance to wasp parasitism has been linked to higher constitutive haemocyte production [69], but this is the first report which demonstrates that the same mechanism could possibly be involved in honey bee resistance to *Varroa* parasitism. Thus, higher cellular immunity appears to be a novel resistance mechanism to *Varroa*, and it is likely contributing, along with the behavioral resistance in the LVG phenotype. In addition, as this is a constitutive trait, future research could examine if LVG bees are also resistant to other pathogens, such as *Spiroplasma melliferum,* where haemocytes are associated with resistance [70] and *Nosema* spp., where haemocytes increase aggregation [71].

Humoral immunity of LVG and HVG bees exposed to *V. destructor* parasitism was estimated from the expression of the AMP immune-related genes, *AmDef-2* and *AmHym-2*. The expression of genes for AMPs in bees are activated by different signaling pathways, which would be the Toll pathway for defensins, and Imd pathway for hymenoptaecins [22,37]. There were no significant differences in the expression of both genes in non-parasitized LVG and HVG bees, but the expression of *AmHym-2* was significantly higher in parasitized LVG than parasitized HVG bees, indicating a triggering of the Imd pathway. However, there was considerable variation in the expression of *AmDef-2*, and it is possible that a significant difference would have been found with additional replications. None of the previous studies selecting bees for LVG [11,12,14,47,49,50,56] examined humoral immunity through gene expression. However, the upregulation of immune-related genes has been observed in studies of adult bees parasitized by *V. destructor*, such as both *AmDef-2* and *AmHym-2* [72,73]. In addition, the expression of *AmDef-1* and *AmHym-1* was higher in parasitized *Varroa*-resistant genotypes compared to susceptible genotypes, indicating that there is genetic variation in defense gene expression among honey bees related to *Varroa* resistance [1,4,74]. Increased expression of AMPs could be a response to stresses, such as cell damage caused by *Varroa* feeding and by DWV replication, or it could contribute to resistance to the parasite. Thus, in addition to hygienic behavior, grooming behavior, and cellular immunity, these results show that humoral immunity appears to be contributing to *Varroa* resistance of LVG bees. However, it is important to examine the expression of more genes regulated by the Toll or Imd pathways, as well as genes related to the JAK/STAT and JNK pathways that also regulate bee immunity [37]. Since AMPs are also important in resistance to a variety of honey bee pathogens, such as *Paenibacillus larvae* [37] and *Spiroplasma melliferum* [70], increased humoral resistance in the LVG bees may also indicate that they have enhanced resistance to a variety of bee diseases.

DWV levels in caged bees of the third generation varied between genotypes, with LVG bees having significantly lower infection levels than HVG bees in both control and *Varroa*-exposed bees. Viral infection levels were highest when bees of both genotypes were challenged with the mite. Clearly, *Varroa* exposure triggers DWV proliferation as has been demonstrated in other studies [1,9]. The only other study that selected bees for LVG that assessed DWV levels was that of Emsen et al. [75], who found 47% lower DWV levels compared with HVG bees, which was very similar to this study. Additionally, bee genotypes with resistance to *Varroa* have been shown to have lower infection levels of DWV than susceptible genotypes [76]. Viral resistance is a part of humoral immunity, and the lower DWV titers observed in bees from *Varroa*-resistant genotypes (selected or natural) compared to susceptible genotypes indicates antiviral mechanisms can differ by bee genotype [1,75,77]. Possible potential mechanisms of viral resistance could be less feeding by *V. destructor* on LVG bees, reducing the amounts of DWV transmitted, increasing resistance to DWV multiplication, such as RNAi [29] or other unknown mechanisms. Penn et al. [78] suggested that RNAi affecting DWV replication does occur but they did not find a strong correlation between *Varroa*-resistant genotypes of honey bees with DWV loads. Thus, humoral immunity as detected by higher AMP gene expression and lower DWV replication are contributing to the LVG phenotype, in addition to higher hygienic behavior, grooming behavior, and cellular immunity.

## 5. Conclusions

Honey bee genotypes selected for low or high *V. destructor* population growth (LVG and HVG, respectively) were examined for multiple potential *Varroa* resistance mechanisms. Rates of hygienic and grooming behaviors in LVG colonies were significantly higher than those in HVG colonies in two of the three generations. For individual bees in the third generation, grooming start time was significantly shorter and the proportion of individuals performing intense grooming higher in the LVG than in the HVG genotype. Cellular immunity was greater as well based on significantly higher haemocyte concentrations in non-parasitized and *Varroa*-parasitized LVG bees. Humoral immunity was greater with *Varroa*-parasitized LVG bees having significantly higher expression of the antimicrobial peptide gene, hymenoptaecin 2. In addition, antiviral resistance may be involved as there were significantly lower levels of DWV in *Varroa*-parasitized LVG bees. While selection for LVG and HVG bees was solely based on *Varroa* population growth, it appears that components of behavioral, cellular, and humoral mechanisms were all selected, potentially contributing to this resistance, rather than just one or two resistance mechanisms. Thus, LVG resistance appears to be a multi-gene trait related to multiple resistance mechanisms. Since all the potential mechanisms examined in this study appeared to contribute to the resistance of LVG bees, future research could examine additional resistance mechanisms as it is possible that even more traits and, thus, more genes are involved. Due to the simplicity of the methodology used to select LVG colonies, based on mite fall, it could be easily implemented by queen breeders. In contrast, selecting for particular resistance mechanisms, such as higher haemocyte concentration or lower virus levels, would be less simple for queen breeders, although this could result in other novel bee genotypes. The approach used in this study has the potential advantage of including multiple resistance mechanisms that could have additive or synergistic effects and be more stable as *Varroa* would have to overcome several modes of resistance.

## Figures and Tables

**Figure 1 insects-16-00385-f001:**
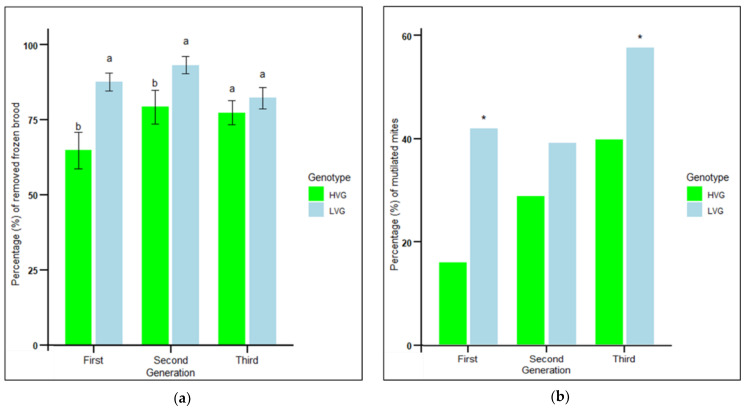
Mean percentage of removed frozen brood ± SEM in 24 h (**a**) and mean percentage of dented and/or mutilated mites recovered from bottom boards (**b**) of colonies during the three generations of selection for high and low *Varroa* growth (HVG and LVG, respectively). Different literals for the percentage of removed frozen brood indicate significant differences between genotypes within each generation based on analysis of variance and Fisher-protected LSD tests of arcsine-square root transformed data (*p* < 0.05). Untransformed values are shown. Asterisks for percentage of dented and/or mutilated mites indicate significant differences between genotypes within each generation based on contingency table analyses with corrected chi-squares (*p* < 0.05).

**Figure 2 insects-16-00385-f002:**
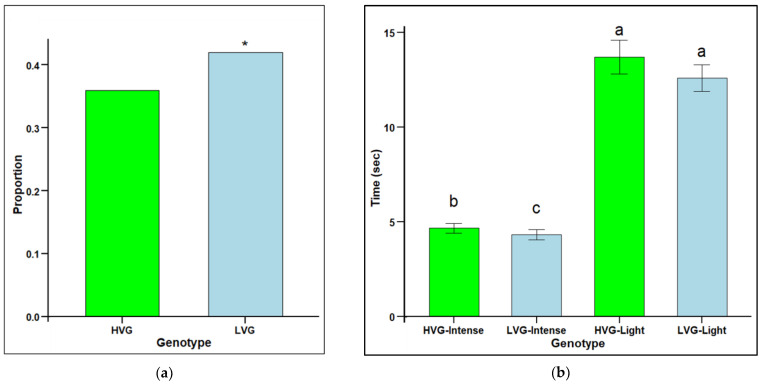
Grooming intensity (proportion of individuals grooming over 180 s (**a**) and grooming starting time (mean time (s) to observe first grooming instances ± SEM) (**b**) for worker honey bees of the third generation of selection for high *Varroa* growth (HVG) and low *Varroa* growth (LVG). Asterisk (*) for the proportion of groomers indicate significant differences between bee types (*p* < 0.05) and are based on contingency table analyses with corrected chi-squares (**a**). Different literals for time to groom indicate significant differences between bee types (*p* < 0.05) and are based on ANOVA and Fisher-protected LSD tests of log-transformed data (**b**). Untransformed values are presented.

**Figure 3 insects-16-00385-f003:**
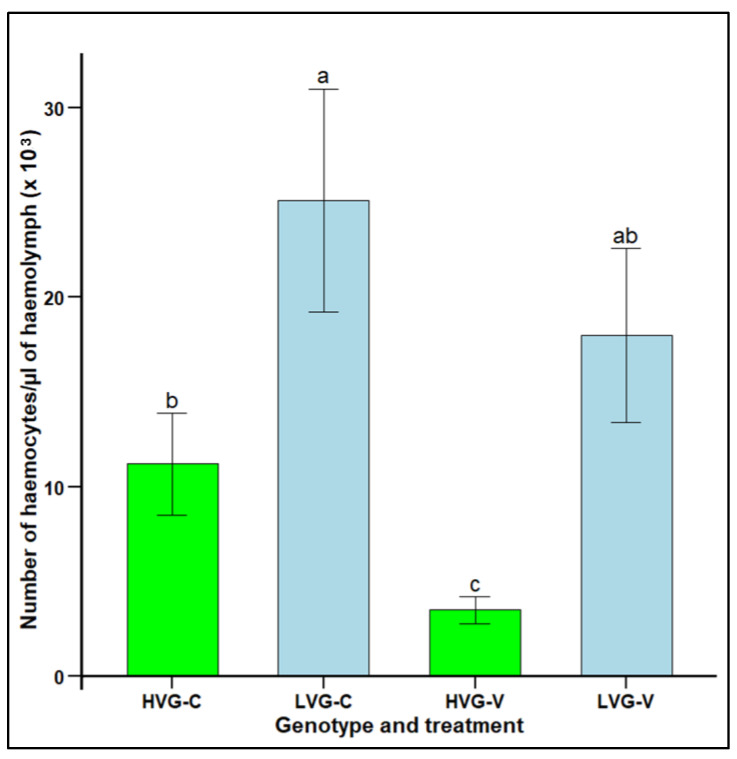
Haemocyte concentration per µL of haemolymph (mean ± SEM) of adult workers from honey bee colonies of the third generation of selection for high and low *Varroa* growth (HVG and LVG, respectively). Bees of both genotypes were exposed (V) or not exposed (C) to *V. destructor* parasitism in cages under a controlled environment. Different literals indicate significant differences between genotypes based on analysis of variance and Fisher-protected LSD tests of log-transformed data. Untransformed values are shown.

**Figure 4 insects-16-00385-f004:**
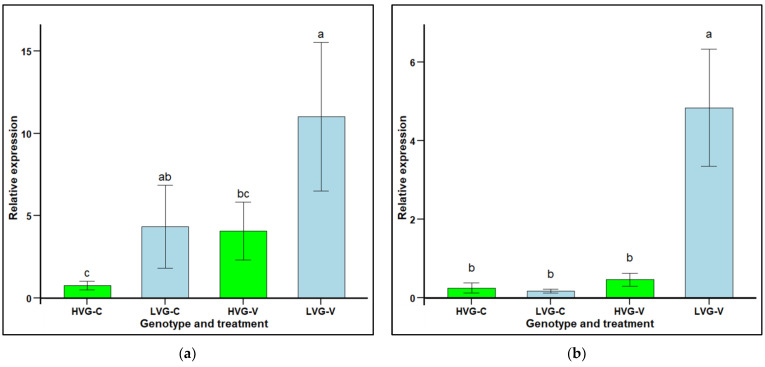
Relative gene expression (RE mean ± SEM) of the immune-related genes, defensin 2 (*AmDef-2*; (**a**) and hymenoptaecin 2 (*AmHym-2*; (**b**), of pooled worker honey bees (n = 15) from six colonies selected for high and six colonies selected for low *Varroa* growth (HVG and LVG, respectively) that were (V) or not (C) challenged with *V. destructor*. The expression of the target genes was calculated using the Livak 22^−ΔΔ^ method, relative to the reference gene, *AmRPS5*, and the non-challenged HVG genotype was used as a calibrator. Different literals represent significant differences based on analyses of variance and Fisher protected LSD tests of log transformed data. Untransformed values are presented.

**Figure 5 insects-16-00385-f005:**
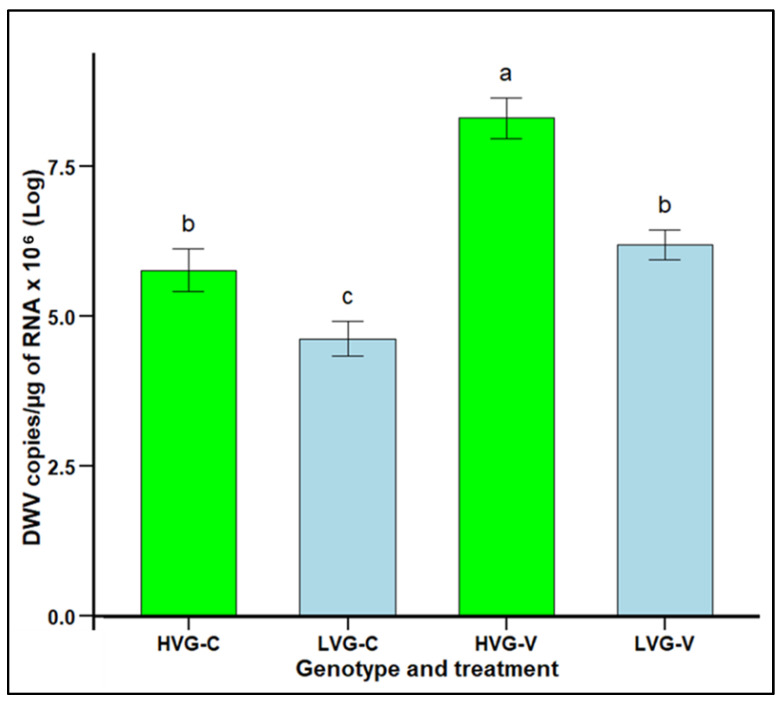
DWV infection levels (mean DWV copies/µg of RNA x 10^6^ ± SEM) of worker honey bees of the third generation of selection for high and low *Varroa* growth (HVG and LVG, respectively) that were exposed (V) or not exposed (C) to *V. destructor*. Different literals indicate significant differences between genotypes based on analysis of variance and Fisher-protected LSD tests of log-transformed data.

## Data Availability

The data from this study will be provided by the corresponding author upon reasonable request. The data are not publicly available due to future publications and collaborations related to the breeding lines.

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
