# Peer review of "Diversity of Potential Resistance Mechanisms in Honey Bees (Apis mellifera) Selected for Low Population Growth of the Parasitic Mite, Varroa destructor"

_insects, 2025, doi:10.3390/insects16040385_

Round 1
Reviewer 1 Report
Comments and Suggestions for Authors
Overall, this publication provides useful information concerning the consequences of a relatively simple selection procedure, helping understand some of the mechanisms involved in bee colony resistance to the Varroa mite.
Some comments and suggestions:
Some careful reading, maybe with help from people not directly involved in the research, could improve the writing. The text is generally well written but still could be improved.
A point that was not addressed is whether these colonies were maintained were treated or not with Varroa control chemicals or other treatment strategies , and with what and when. Maybe this is explained in previous publications concerning this selection work, but it should be included in this paper.
Also, were the three selection cycles done in different years?
“which reduces the longevity of bees both for adults without causing body deformations, and brood with deformed body and wings” – this text could be improved. Reducing longevity of brood does not make sense here.
“They also promote wound healing and haemolymph clotting after V. destructor attack [24], and they also produce anti-viral compounds” – this is awkward sounding text. “They also twice” in the same sentence.
“Samples of 10 treated bees of each genotype that had or had not been exposed to V. destructor were collected at eight days post-treatment” the use of the term treatment is not completely clear to this reader. Does this mean eight days after the bees were infested?
“Additionally, Varroa parasitized bees of the two genotypes were assessed for DWV levels to determine if antiviral resistance was involved.” Less DWV virus could be a consequence of less Varroa instead of antiviral resistance.
“Briefly, a metal cylinder (4.12 cm diameter)” - inner diameter?
“The number of mites with mutilated legs and dents on the idiosoma were counted” and/or dents? Or only mites that had both?
“The time (s) taken by each bee to start performing the first grooming instance was measured for up to 3 min using a stopwatch with a resolution of 1/100 s and an accuracy of 0.001%” Considering that the time registered apparently depends on the reaction time of the person operating the stopwatch, this description of resolution and accuracy does not appear to be relevant. Maybe the model of this Fisherbrand stopwatch could be included instead.
“For Varroa exposure, bees were artificially parasitized with one mite per individual by placing a mite with a fine paintbrush onto each bee through the cage screen” here the origin of the mites used should be explained.
“65% sugar syrup” what sugar - sucrose? %weight/weight?
“(65% sugar syrup mixed with powdered sugar)” powdered sucrose? Ratio of powdered sugar to syrup or explain consistency
“by placing clean? sponges with dsH2O on the cage screen” pieces of sponge? Normal kitchen sponges?
“Samples of 10 treated bees of each genotype that had or had not been exposed to V. destructor were collected at eight days post-treatment. Each bee was held manually with the index and thumb fingers to expose the dorsal part of the insect’s abdomen” bees were anesthetized to do this? How?
“Calibration curves to convert Ct values to DWV genome copies (gc) were done using a 300 bp gBlocks” I think that it would make more sense to say: using 300 bp gBlocks – leaving out the “a”
“Hygienic behavior estimated from the percent frozen brood removed by bees of the LVG colonies was significantly higher than that of the HVG colonies in the first and second generations, but not in the third generation (F1,97= 15.25, p<0.001)” Do the F value and p include the third generation?
“Thus, it is possible that different sets of grooming behavior genes may have been more greatly selected in certain generations.” more strongly selected – in each of the generations
“This behavior may reduce mite fertility and fecundity because mite reproduction and mortality of mite offspring is higher in recapped cells than in mite-infested cells that are not opened [58].” Would be useful to explain why this makes sense.
“For individual bees in the third generation, grooming start time was significantly shorter and grooming intensity more frequent in the LVG than in the HVG genotype” what is meant here by “grooming intensity more frequent”? what is a more frequent grooming intensity?
“Mite damage rates from LVG colonies in the first and third generations were 61% and 30% greater than those of HVG colonies, respectively, which was associated with 47% and 66% lower Varroa infestation levels in adult bees and brood, respectively.” Affirmations such as the above sentence in the Discussion should be followed by an indication of where the data can be found.
“The approach used in this study has the potential advantage of including multiple resistance mechanisms that could have additive or synergistic effects and be more stable as Varroa would have to overcome several modes of resistance.” This could be explained better since lower DWV titers would be useful but should not be included as a mode of resistance of the mites.
“Thus, LVG resistance appears to be a multi-gene rather than a single gene trait” Why mention a single gene trait in the conclusions? I have not seen any published work that postulates Varroa resistance as a single gene trait.
The highly cited publication by Buchler et al 2010 concerning Breeding for resistance against Varroa in Europe, cited in this manuscript, reference 14, states that mite fall, damaged mites, and hygienic behavior were not found to be correlated with survival of Varroa infested colonies. A discussion of the differences in the value of these parameters for selection for Varroa resistance could be useful in this work. Various other publications have shown that the percent damaged mites found on the bottom board did not differ between bee populations that are resistant to Varroa versus susceptible bees with high infestation rates. Given that these parameters are the basis of the selection method used and recommended by the authors, these discrepancies could be included in the Discussion and/or Introduction.
In the conclusions: “Due to the simplicity of the methodology used to select LVG colonies, it could be easily implemented by queen breeders.”, it would be useful to include “based on mite fall” after “select LVG colonies”
The affirmations that hygienic behavior and grooming were greater in the LVG bees compared to the HVG bees were not supported in all generations. This is not reflected in the statements made in the abstract and the conclusions. Authors should make this clear or justify their statements in this regard.
Author Response
Comment 1: Some careful reading, maybe with help from people not directly involved in the research, could improve the writing. The text is generally well written but still could be improved.
Response 1: Several modifications in the text have been made. Including the title (lines 2-4).
Comment 2: A point that was not addressed is whether these colonies were maintained were treated or not with Varroa control chemicals or other treatment strategies, and with what and when. Maybe this is explained in previous publications concerning this selection work, but it should be included in this paper.”
Response 2: Edited in the new version of the manuscript (lines 107-109). “Varroa treatments were applied in colonies in fall of each year using amitraz (Apivar, Veto-Pharma, Saint-Benoit-du-Sault, 36310 Chaillac, France Chaillac, FR) following the manufacture’s instructions.”
Comment 3: Also, were the three selection cycles done in different years?
Response 3: Edited in the new version of the manuscript (line 106). “Each generation was one year”.
Comment 4: “which reduces the longevity of bees both for adults without causing body deformations, and brood with deformed body and wings” – this text could be improved. Reducing longevity of brood does not make sense here.
Response 4: Edited in the new version of the manuscript (line 59-61).
Comment 5:“They also promote wound healing and haemolymph clotting after V. destructor attack [24], and they also produce anti-viral compounds” – this is awkward sounding text. “They also twice” in the same sentence.
Response 5: Edited in the new version of the manuscript (line 83).
Comment 6: “Samples of 10 treated bees of each genotype that had or had not been exposed to V. destructor were collected at eight days post-treatment” the use of the term treatment is not completely clear to this reader. Does this mean eight days after the bees were infested?
Response 6: Edited in the new version of the manuscript (lines 172-174). Yes, we mean after being infested.
Comment 7: “Additionally, Varroa parasitized bees of the two genotypes were assessed for DWV levels to determine if antiviral resistance was involved.” Less DWV virus could be a consequence of less Varroa instead of antiviral resistance.
Response 7: Edited in the new version of the manuscript (lines 95-97). “Additionally, Varroa parasitized bees of the two genotypes were assessed for DWV levels to determine if antiviral resistance was involved for bees individually parasitized by the mite.”
Comment 8: “Briefly, a metal cylinder (4.12 cm diameter)” - inner diameter?
Response 8: Edited in the new version of the manuscript (lines 113-114). Internal diameter is now reported.
Comment 9: “The number of mites with mutilated legs and dents on the idiosoma were counted” and/or dents? Or only mites that had both?
Response 9: Edited in the new version of the manuscript (lines 121, 125, 281, 297, 302, 406 and 430). “The number of mites with mutilated legs and/or dents on the idiosoma were counted.”
Comment 10:“The time (s) taken by each bee to start performing the first grooming instance was measured for up to 3 min using a stopwatch with a resolution of 1/100 s and an accuracy of 0.001%” Considering that the time registered apparently depends on the reaction time of the person operating the stopwatch, this description of resolution and accuracy does not appear to be relevant. Maybe the model of this Fisherbrand stopwatch could be included instead.
Response 10: Edited in the new version of the manuscript (lines 139-143). “The time (s) taken by each bee to start performing the first grooming instance was measured for up to 3 min using a stopwatch with a resolution of 1/100 s and an accuracy of 0.001% (Catalog number 06-662-56, Fisherbrand, Fisher Scientific, Mississauga, ON, CA). This was done as a blind test to the observer.”
Comment 11: “For Varroa exposure, bees were artificially parasitized with one mite per individual by placing a mite with a fine paintbrush onto each bee through the cage screen” here the origin of the mites used should be explained.
Response 11: This information was provided in lines 150-152: “To obtain female Varroa, frames covered with adult bees from highly infested colonies that were unrelated to the experimental bees…”
Comment 12: “65% sugar syrup” what sugar - sucrose? %weight/weight?
Response 12: Edited in the new version of the manuscript (line 1669. “65% sucrose syrup…”
Comment 13: “(65% sugar syrup mixed with powdered sugar)” powdered sucrose? Ratio of powdered sugar to syrup or explain consistency
Response 13: Edited in the new version of the manuscript regarding consistency (line 170).
Comment 14:“by placing clean? sponges with dsH2O on the cage screen” pieces of sponge? Normal kitchen sponges?
Response 14: Edited in the new version of the manuscript (line 171-172). “by placing a clean 2 × 2 cm sponge piece (Scotch Brite, 3M, Two Harbors, MN, USA) with dsH2O on the cage screen.”
Comment 15: “Samples of 10 treated bees of each genotype that had or had not been exposed to V. destructor were collected at eight days post-treatment. Each bee was held manually with the index and thumb fingers to expose the dorsal part of the insect’s abdomen” bees were anesthetized to do this? How?
Response 15: Bees were not anesthetized, but the text was edited in the new version of the manuscript (lines 172-175). “Samples of 10 bees of each genotype that had or had not been exposed to V. destructor were collected at eight days post-exposure to the mite. Each bee was collected directly from the cage, and held manually with the index and thumb fingers to expose the dorsal part of the insect’s abdomen.”
Comment 16: “Calibration curves to convert Ct values to DWV genome copies (gc) were done using a 300 bp gBlocks” I think that it would make more sense to say: using 300 bp gBlocks – leaving out the “a”
Response 16: Edited in the new version of the manuscript (line 257). “Calibration curves to convert Ct values to DWV genome copies (gc) were done using 300 bp gBlocks …”
Comments 17: “Hygienic behavior estimated from the percent frozen brood removed by bees of the LVG colonies was significantly higher than that of the HVG colonies in the first and second generations, but not in the third generation (F1,97= 15.25, p<0.001)” Do the F value and p include the third generation?
Response 17: Yes, it does.
Comment 18: “Thus, it is possible that different sets of grooming behavior genes may have been more greatly selected in certain generations.” more strongly selected – in each of the generations
Response 18: Edited the new version of the manuscript (line 427). “Thus, it is possible that different sets of grooming behavior genes may have been more strongly selected in certain generations.”
Comment 19: “This behavior may reduce mite fertility and fecundity because mite reproduction and mortality of mite offspring is higher in recapped cells than in mite-infested cells that are not opened [58].” Would be useful to explain why this makes sense.
Response 19: Edited in the new version of the manuscript (line 469-470). Sentence was removed.
Comment 20: “For individual bees in the third generation, grooming start time was significantly shorter and grooming intensity more frequent in the LVG than in the HVG genotype” what is meant here by “grooming intensity more frequent”? what is a more frequent grooming intensity?
Response 20: Edited in the new version of the manuscript (lines 580-583). “For individual bees in the third generation, grooming start time was significantly shorter and the proportion of individuals performing intense grooming higher in the LVG than in the HVG genotype.”
Comment 21: “Mite damage rates from LVG colonies in the first and third generations were 61% and 30% greater than those of HVG colonies, respectively, which was associated with 47% and 66% lower Varroa infestation levels in adult bees and brood, respectively.” Affirmations such as the above sentence in the Discussion should be followed by an indication of where the data can be found.
Response 21: Edited in the new version of the manuscript (line 284-288). “Mite damage rates from LVG colonies in the first and third generations were 61% and 30% greater than those of HVG colonies (Figure 1b), respectively, which was associated with 47% and 66% lower Varroa infestation levels in adult bees and brood, respectively, as reported previously [12].”
Comment 22: “The approach used in this study has the potential advantage of including multiple resistance mechanisms that could have additive or synergistic effects and be more stable as Varroa would have to overcome several modes of resistance.” This could be explained better since lower DWV titers would be useful but should not be included as a mode of resistance of the mites.
Response 22: Lower DWV titers in Varroa parasitized bees may indicate that there are antiviral resistance mechanisms in LVG bees than HVG, after bees from both genotypes were parasitized with the same number of mites, and for the same period of time.
Comment 23: “Thus, LVG resistance appears to be a multi-gene rather than a single gene trait” Why mention a single gene trait in the conclusions? I have not seen any published work that postulates Varroa resistance as a single gene trait.
Response 23: Edited in the new version of the manuscript (line 588-596). “While selection for LVG and HVG bees was solely based on Varroa population growth, it appears that components of behavioral, cellular and humoral mechanisms were all selected, potentially contributing to this resistance, rather than just one or two resistance mechanisms. Thus, LVG resistance appears to be a multi-gene trait related to multiple resistance mechanisms. Since all the potential mechanisms examined in this study appeared to contribute to resistance of LVG bees, future research could examine additional resistance mechanisms as it is possible that even more traits and thus more genes are involved.”
Comment 24: The highly cited publication by Buchler et al 2010 concerning Breeding for resistance against Varroa in Europe, cited in this manuscript, reference 14, states that mite fall, damaged mites, and hygienic behavior were not found to be correlated with survival of Varroa infested colonies. A discussion of the differences in the value of these parameters for selection for Varroa resistance could be useful in this work. Various other publications have shown that the percent damaged mites found on the bottom board did not differ between bee populations that are resistant to Varroa versus susceptible bees with high infestation rates. Given that these parameters are the basis of the selection method used and recommended by the authors, these discrepancies could be included in the Discussion and/or Introduction.
Response 24: It is true that some studies have not found these correlations, but it is also true that the relationship between mite damage and hygienic behavior with colony health and survivorship has been demonstrated in other studies not conducted in Europe. Therefore, we added the following text to the Discussion (lines 452-463): “Studies conducted in Europe did not find a correlation of mite fall, damaged mites, or hygienic behavior, to survival of Varroa infested colonies [14]. However, other studies conducted in the Americas and the Middle East have shown a relationship of these traits to colony health and survivorship [16, 17, 31, 45]. Perhaps the relative contribution of different traits to the resistance of bees to Varroa mites and their effect on colony survivorship varies depending upon the environment or some other factor. Therefore, more studies conducted in different regions of the world are needed to determine whether these traits are useful to increase colony survivorship in different environments.”
Comment 25: In the conclusions: “Due to the simplicity of the methodology used to select LVG colonies, it could be easily implemented by queen breeders.”, it would be useful to include “based on mite fall” after “select LVG colonies”
Response 25: Edited in the new version of the manuscript (line 596-598). “Due to the simplicity of the methodology used to select LVG colonies, based on mite fall, it could be easily implemented by queen breeders.”
Comment 26: The affirmations that hygienic behavior and grooming were greater in the LVG bees compared to the HVG bees were not supported in all generations. This is not reflected in the statements made in the abstract and the conclusions. Authors should make this clear or justify their statements in this regard.
Response 26: To correct this, a sentence in conclusions (lines 579-580) changed to “Rates of hygienic and grooming behaviors in LVG colonies were significantly higher than those in HVG colonies in two of the three generations.”, and a sentence in the abstract (line 32-35) changed to “Hygienic and grooming behavior rates in LVG colonies were significantly higher than those in HVG colonies for two out of three generations of selection, indicating that behavioral resistance to the mite was increased.”
Reviewer 2 Report
Comments and Suggestions for Authors
My ultimate take away is that the authors, in fact, do not show evidence of diverse mechanisms of varroa resistance. They present data on one bona fide mechanism of varroa resistance - grooming behavior. Then, they find that the varroa resistance phenotype is correlated with higher hemocyte concentrations an higher AMP expression, but they provide no evidence that these are mechanistically linked to varroa resistance. I'm sure if the authors sequenced their stocks they would find many genetic variants differing between the two populations with no causal link to the observed phenotype. I think the data are relatively interesting, but in my opinion should be packaged into a paper with a title which more accurately reflects the findings like "Immune correlates in a varroa-resistant stock selection program". However, I think there are also data and discussion missing that makes this unpublishable at the current moment. I have some more comments below:
1. Why were intermediate groomers removed? These should be included or a better justification should be supplied. This might change the reading of the data significantly – were LVG bees higher in intermediate behaviors?
2. I am not convinced that grooming behavior isn’t consistently linked with lower varroa, which is stated in the discussion – I cannot locate reference 47 since the DOI leads to a page saying “DOI not found”.
Emsen, B.; Petukhova, T.; Guzman-Novoa, E. Factors limiting the growth of Varroa destructor populations in selected honey bee 696 (Apis mellifera L.) colonies. JAVA 2012, 11, 4519–4525. https://doi.org/10.3923/javaa.2012.4519.4525.
Plus they undermine their own argument by supposedly finding indirect evidence of higher grooming in the form of mutilated mites
3. When they exposed LVG and HVG bees to varroa did the authors measure varroa mortality? Or assess whether the bees were parasitized at the time of collection? It's not clear what the relevance is of the data without an assessment of parasitization during the experiment and at the time of collection.
4. What even is the proposed mechanism for higher hemocyte concentration and lower mites? The authors mention that hemocytes produce ROS and that hemocytes are associated with resistance to endoparasitoids of fruit flies, but I do not immediately see the logical connection to suppressing mite reproduction since they look in adult bees. Could hemocytes produce such a high level of ROS that a mere 1.5X increase would have toxic effects? I think attempting to link these two things requires experiments in larvae or pupae where a higher hemocyte concentration might have a clear effect on mite reproduction since the matriarch mite establishes a wound for feeding - ergo, enhanced wound healing may stem mite feeding..
5. Line 431-432: “In this study, 14% more LVG bees performed grooming events 431 within 3 min of a stimulus than HVG bees.”
Where did this number come from? I see an increase from 0.36 to 0.42 which is 8%
6. After reading the entire discussion a few times I am still unclear what mechanisms the authors believe are limiting varroa reproduction and why. They argue that their grooming rates are not much higher in LVG versus HVG, so it must be due to hemocytes and AMPs but they provide no sound literature-based support for this argument. Perhaps there is better literature out there that supports their conclusions.
Author Response
Comment: My ultimate take away is that the authors, in fact, do not show evidence of diverse mechanisms of varroa resistance. They present data on one bona fide mechanism of varroa resistance - grooming behavior. Then, they find that the varroa resistance phenotype is correlated with higher hemocyte concentrations an higher AMP expression, but they provide no evidence that these are mechanistically linked to varroa resistance. I'm sure if the authors sequenced their stocks they would find many genetic variants differing between the two populations with no causal link to the observed phenotype. I think the data are relatively interesting, but in my opinion should be packaged into a paper with a title which more accurately reflects the findings like "Immune correlates in a varroa-resistant stock selection program". However, I think there are also data and discussion missing that makes this unpublishable at the current moment. I have some more comments below:
Response: These are all potential resistance mechanisms showing the diversity of differences between HVG and LVG bees, and we have modified the title of the paper to reflect that. This work is part of the characterization of these bees, as has been done in other papers selecting for Varroa resistance. The strength of this work is that more potential mechanisms were examined than typically reported in papers characterizing bees selected for Varroa resistance. We have not examined every potential mechanism or assessed the relative contribution of each, but we do state that other mechanisms could be involved, and individual potential mechanisms could be assessed by direct selection in the future. While grooming and hygienic behaviors have been more studied in bee breeding than other potential mechanisms, that is a reflection of the types of studies rather than a definitive statement as to their contribution to resistance, and other potential resistance mechanisms could have been indirectly selected and contributing to their resistance in addition to the change in behavior.
Comment 1. Why were intermediate groomers removed? These should be included or a better justification should be supplied. This might change the reading of the data significantly – were LVG bees higher in intermediate behaviors?
Response 1. Intermediate groomers were removed as the assessment was based on reference [15].
Comment 2. I am not convinced that grooming behavior isn’t consistently linked with lower varroa, which is stated in the discussion – I cannot locate reference 47 since the DOI leads to a page saying “DOI not found”.
Emsen, B.; Petukhova, T.; Guzman-Novoa, E. Factors limiting the growth of Varroa destructor populations in selected honey bee 696 (Apis mellifera L.) colonies. JAVA 2012, 11, 4519–4525. https://doi.org/10.3923/javaa.2012.4519.4525.
Plus they undermine their own argument by supposedly finding indirect evidence of higher grooming in the form of mutilated mites
Response 2. The statement that grooming behavior is not consistently linked with lower Varroa is directly based on our results shown in Figure 1b, where two of the three generations showed significantly higher grooming behaviour in LVG bees. As stated in the discussion (lines 407-409), “Mite damage rate in LVG colonies was significantly higher than that in HVG colonies in the first and third generations, but not in the second generation, and thus, it was not consistently associated with low Varroa growth”. Thus, this is a statement about our results, not an opinion. Those results were additionally mentioned in the last two sentences of the same paragraph (lines 417-422) to infer that other mechanisms may contribute to Varroa resistance in LVG bees, in addition to hygienic and grooming behavior.
Comment 3. When they exposed LVG and HVG bees to varroa did the authors measure varroa mortality? Or assess whether the bees were parasitized at the time of collection? It's not clear what the relevance is of the data without an assessment of parasitization during the experiment and at the time of collection.
Response 3. Bee mortality on Varroa parasitized bees was previously published in Figure 5 at De La Mora, A.; Goodwin, P. H.; Emsen, B.; Kelly, P. G.; Petukhova, T.; Guzman-Novoa, E. Selection of honey bee (Apis mellifera) genotypes for three generations of low and high population growth of the mite Varroa destructor. Animals 2024, 14, 3537. https://doi.org/10.3390/ani14233537.”
Comment 4. What even is the proposed mechanism for higher hemocyte concentration and lower mites? The authors mention that hemocytes produce ROS and that hemocytes are associated with resistance to endoparasitoids of fruit flies, but I do not immediately see the logical connection to suppressing mite reproduction since they look in adult bees. Could hemocytes produce such a high level of ROS that a mere 1.5X increase would have toxic effects? I think attempting to link these two things requires experiments in larvae or pupae where a higher hemocyte concentration might have a clear effect on mite reproduction since the matriarch mite establishes a wound for feeding - ergo, enhanced wound healing may stem mite feeding.
Response 4. Edits are now included in the new version of the manuscript (Lines 493-500): “Cellular immunity, based on the concentration of haemocytes in the haemolymph, was significantly higher in LVG than in HVG bees both with and without Varroa parasitism. As increased cellular immunity has never been related to Varroa resistance, this was unexpected. Because newly emerged bees were determined to not have Varroa parasitism prior to being placed in cages in this study, it could be proposed that increased haemocyte concentrations in bees without Varroa shows that a basal function has been indirectly selected during the development of LVG colonies.” (Lines 521-524): “Thus, higher cellular immunity appears to be a novel resistance mechanism to Varroa, and it is likely contributing along with the behavioral resistance in the LVG phenotype.”
Comment 5. Line 431-432: “In this study, 14% more LVG bees performed grooming events 431 within 3 min of a stimulus than HVG bees.”
Where did this number come from? I see an increase from 0.36 to 0.42 which is 8%
Response 5. The value of 14% came from dividing 0.36 by 0.42, which result is 0.857. Then, multiplying 0.857 by 100, which result is 85.7%. Finally, subtracting 85.7% from 100%, to obtain 14%.
Comment 6. After reading the entire discussion a few times I am still unclear what mechanisms the authors believe are limiting varroa reproduction and why. They argue that their grooming rates are not much higher in LVG versus HVG, so it must be due to hemocytes and AMPs but they provide no sound literature-based support for this argument. Perhaps there is better literature out there that supports their conclusions.
Response 6. We have revised the discussion in several places to make it clearer that all the mechanisms examined may be contributing to resistance, which is why our conclusion is that (lines 589-596) “it appears that components of behavioral, cellular and humoral mechanisms were all selected, potentially contributing to this resistance, rather than just one or two resistance mechanisms. Thus, LVG resistance appears to be a multi-gene trait related to multiple resistance mechanisms. Since all the potential mechanisms examined in this study appeared to contribute to resistance of LVG bees, future research could examine additional resistance mechanisms as it is possible that even more traits and thus more genes are involved.”. Extensive revisions have been made throughout the discussion as well as the abstract to make this clearer.
Round 2
Reviewer 2 Report
Comments and Suggestions for Authors
Thank you to the authors for the work they put into improving the manuscript.